# High Stretch Modulates cAMP/ATP Level in Association with Purine Metabolism via miRNA–mRNA Interactions in Cultured Human Airway Smooth Muscle Cells

**DOI:** 10.3390/cells13020110

**Published:** 2024-01-05

**Authors:** Mingzhi Luo, Chunhong Wang, Jia Guo, Kang Wen, Chongxin Yang, Kai Ni, Lei Liu, Yan Pan, Jingjing Li, Linhong Deng

**Affiliations:** Changzhou Key Laboratory of Respiratory Medical Engineering, Institute of Biomedical Engineering and Health Sciences, School of Medical and Health Engineering, Changzhou University, Changzhou 213164, China19085235275@smail.cczu.edu.cn (K.W.);

**Keywords:** high stretch, airway smooth muscle cells, ventilator-induced lung injury, microRNA, purine metabolism

## Abstract

High stretch (>10% strain) of airway smooth muscle cells (ASMCs) due to mechanical ventilation (MV) is postulated to contribute to ventilator-induced lung injury (VILI), but the underlying mechanisms remain largely unknown. We hypothesized that ASMCs may respond to high stretch via regulatory miRNA–mRNA interactions, and thus we aimed to identify high stretch-responsive cellular events and related regulating miRNA–mRNA interactions in cultured human ASMCs with/without high stretch. RNA-Seq analysis of whole genome-wide miRNAs revealed 12 miRNAs differentially expressed (DE) in response to high stretch (7 up and 5 down, fold change >2), which target 283 DE-mRNAs as identified by a parallel mRNA sequencing and bioinformatics analysis. The KEGG and GO analysis further indicated that purine metabolism was the first enriched event in the cells during high stretch, which was linked to miR-370-5p–PDE4D/AK7. Since PDE4D/AK7 have been previously linked to cAMP/ATP metabolism in lung diseases and now to miR-370-5p in ASMCs, we thus evaluated the effect of high stretch on the cAMP/ATP level inside ASMCs. The results demonstrated that high stretch modulated the cAMP/ATP levels inside ASMCs, which could be largely abolished by miR-370-5p mimics. Together, these findings indicate that miR-370-5p–PDE4D/AK7 mediated high stretch-induced modulation of cAMP and ATP synthesis inside ASMCs. Furthermore, such interactive miRNA–mRNA pairs may provide new insights for the discovery of effective biomarkers/therapeutic targets for the diagnosis and treatment of VILI and other MV-associated respiratory diseases.

## 1. Introduction

Mechanical ventilation (MV) with a large breathing volume or high pressure is clinically critical for providing life support to patients suffering from acute respiratory distress syndrome (ARDS). However, such unphysiological breathing volume or pressure can over-distend the lung and cause a high stretch (>10% strain) of the airways, which has been postulated to aggregate lung injury, a phenomenon known as ventilator-induced lung injury (VILI) [1,2,3,4,5]. Unfortunately, there is currently no effective pharmacological intervention for VILI, and the consequence of uncontrolled VILI can be high mortality (e.g., up to 60% during the COVID-19 pandemic) [6]. Therefore, there is an urgent need to fully elucidate the pathological mechanisms of VILI in order to improve the outcome of MV treatment [7,8,9,10].

Clinical studies of patients under MV for over six days indicated decreased lung compliance and increased airway inflammation [6,11,12,13,14,15], reflecting pathological alterations in respiratory structure and function [16,17], particularly, aberrant behaviors of the airway smooth muscle cells (ASMCs) [18,19,20,21,22]. Since ASMCs are the primary mechanosensitive cells in the airway system and are known to change phenotype in response to stretch [22], alterations of ASMCs subjected to high stretch during MV are likely to play a central role in the underlying pathological mechanisms and thus may provide potential targets for prevention of VILI.

Indeed, several pharmacological targets of ASMCs have been explored previously, including the blockade of transcription factors and neutralization of inflammation, but all failed in the attempt to effectively reduce the mortality due to VILI [7,8,9,23]. More recently, noncanonical molecular pathways, such as microRNAs (miRNAs)-mRNA interaction networks, have emerged as novel potential targets to mediate the mechanoresponse of ASMCs [24]. For example, miRNA let-7a has been demonstrated to not only modulate mechanically associated genes in melanoma cells [25] but also in biomechanical and inflammatory phenotypes of the ASMCs [26]. It has also been identified that TRPV4 and ERK1/2 are involved in the stretch-induced modulation of inflammatory factors in ASMCs [27,28]. However, these conventional hypothesis-driven studies are insufficient to fully reveal the stretch-induced prominent responses from an integral perspective. This limitation can be overcome with emerging high-throughput screening techniques, such as whole genome-wide RNA sequencing (RNA-Seq) [29,30]. Through bioinformatic analysis of genes derived from RNA-Seq, it is possible to obtain tremendous gene expression information and explore disease-associated genes and their biological functions without the presumptions of any particular hypothesis [31].

Inspired by these developments, we thus sought to systematically identify miRNA and mRNA expression profiles and their specific interactions in ASMCs whether or not subjected to high stretch. In doing so, we hope to reveal the full regulatory network of the high stretch-induced behavioral changes of ASMCs so that a potential new target or approach may be discovered for effective prevention or treatment of VILI [32]. Using RNA sequencing and bioinformatics analysis, we identified 12 miRNAs (7 up-regulated and 5 down-regulated) and 283 targeted mRNAs (141 up-regulated and 142 down-regulated) that were differently expressed in ASMCs cultured in static and high-stretch conditions. However, it was unexpectedly identified that purine metabolism was the top enriched event in the ASMCs exposed to high stretch, which was further linked to 7 miRNA–mRNA interactive pairs, including miR-370-5p–PDE4D/AK7. Since PDE4D/AK7 has been previously linked to cAMP/ATP metabolism in lung diseases [33,34] and now to miR-370-5p in ASMCs, we thus evaluated the effect of high stretch on the cAMP/ATP level inside ASMCs. The results show that high stretch modulated cAMP/ATP levels inside ASMCs, which could be largely abolished by miR-370-5p. Additionally, our study revealed purine metabolism and novel miRNA–mRNA interactions for the response of ASMCs to high stretch, which provides a scientific basis for targeting purine metabolism in treatment/prevention of VILI.

## 2. Materials and Methods

### 2.1. Materials

Transferrin (#T8158) and insulin (#91077C) were purchased from Sigma-Aldrich (St. Louis, MO, USA). Collagen type I (#08-115) was purchased from Advanced BioMatrix (#5279, Poway, CA, USA). Dulbecco’s modified Eagle’s medium (#11885092, DMEM), fetal bovine serum (FBS, #16000-044), penicillin-streptomycin (#15140122), and trypsin (#25200056) were purchased from Thermo Fisher Scientific (Waltham, MA, USA). Cell culture flasks (#3073) and plates (#CLS3506) were purchased from Corning Incorporated (Corning, NY, USA). All other reagents were purchased from Fisher Scientific unless noted otherwise.

### 2.2. Human ASMCs Cultured with/without High Stretch

Primary human ASMCs (#BNCC339826) were purchased from the BeNa Culture Collection (Beijing, China) and used in passages 3–10 which maintained normal “hill and valley” morphology and cellular functions, including physiological response to contractile/relaxant agonist and α-smooth muscle actin (SMA) expression (#ab119952, Abcam, Waltham, MA, USA) [26,35,36,37]. These cells were cultured in DMEM supplemented with 10% FBS, 2 mM L-glutamine, 100 units/mL penicillin, and 100 μg/mL streptomycin in an incubator containing 5% CO_2_ humidified at 37 °C according to the method described previously. Before the experiment, exponential proliferating ASMCs (2 × 10^4^ cells/cm^2^) were plated on the type I collagen-coated elastic membrane of Bioflex 6-well plates (FlexCell International, Hillsborough, NC, USA) followed by 24 h serum deprivation with transferrin (5 ng/mL) and insulin (5 ng/mL). Subsequently, a 13% cyclic strain in sinusoidal waveform (0.5 Hz, 1 s deformation alternating with 1 s relaxation) was applied to the ASMCs for 72 h using Flexcell 5000 (FlexCell International), which simulated the in vivo high stretch on ASMCs in ARDS during MV as estimated according to the literature reports [3,4,26,37,38,39]. The ASMCs prepared in the same way but maintained in static condition for 72 h were used as controls. After high-stretch treatment, RNAs were collected from the ASMCs and analyzed for the responsive events and signaling molecules, using whole genome-wide RNA-Seq or quantitative PCR (qPCR), bioinformatics analysis, and function identification. This study protocol is schematically shown in Figure 1.

### 2.3. Whole Genome-Wide RNA Sequencing of Cultured ASMCs

Total RNA was extracted from cultured ASMCs using TRI Reagent RNA Isolation Reagent (#T9424, Sigma-Aldrich, St. Louis, MO, USA). Then the RNA samples were sent to Shanghai Majorbio Bio-Pharm Technology Co. (Shanghai, China) for quality test, library construction, and sequencing of mRNA and miRNA on the Illumina NovaSeq 6000 and HiSeq^TM^ 2000 platform, respectively (*n* = 3, per group) [40,41,42].

### 2.4. Analysis of Differentially Expressed mRNAs and miRNAs in ASMCs

More specifically, the RNA-Seq data was analyzed to identify the differentially expressed mRNAs and miRNAs in ASMCs cultured in either static or high-stretch conditions on the Majorbio Cloud Platform (Version 2.0, https://www.majorbio.com/, accessed on 20 February 2022) [42,43]. The mRNA or miRNA abundance was quantified as transcripts per million (TPM), and those of low abundance (TPM < 10) were filtered out. Then the differential expression of mRNAs or miRNAs due to high stretch was determined using DESeq2, with *p* values calculated using the Benjamini–Hochberg model. A 2-fold change was used as the cut-off value (*p* < 0.05) to identify the mRNAs and miRNAs that were differentially expressed (labeled as DE-mRNAs or DE-miRNAs, respectively) in ASMCs when cultured with high stretch as compared to those in static condition [44].

The DE-mRNAs that are target mRNAs of the DE-miRNAs in ASMCs treated with high stretch were further predicted by miRanda (Version 3.3a, http://www.miranda.org/, accessed on 25 September 2022), TargetScan (Version 7.0, http://www.targetscan.org/, accessed on 25 September 2022), and RNAhybrid (Version 2.1.2, http://bibiserv.techfak.uni-bielefeld.de/rnahybrid/, accessed on 25 September2022). Those genes overlapping the target mRNAs of down/up-regulated DE-miRNAs with up/down-regulated DE-mRNAs were identified as target DE-mRNAs.

### 2.5. Bioinformatics Analysis

Once the target DE-mRNAs were identified, the DE-mRNAs-associated gene functions that were enriched in response to high stretch were further evaluated with Kyoto Encyclopedia of Genes and Genomes (KEGG) pathway analysis and Gene Ontology (GO) analysis, which included biological processes (BP) and molecular functions (MF) performed by the Database for Annotation, Visualization, and Integrated Discovery database (DAVID, Version 6.8, https://david.ncifcrf.gov/, accessed on 12 October 2022), respectively, and false discovery rate (FDR) < 0.05 was considered significant [45].

### 2.6. Analysis of Protein–Protein Interaction (PPI)

The protein–protein interaction (PPI) between DE-mRNAs was constructed using the Search Tool for the Retrieval of Interacting Genes/Proteins (STRING 11.5, https://string-db.org/, accessed on 25 October 2022) with a confidence score ≥ 0.70 [46]. A visual PPI network was constructed using Cytoscape software (Version 3.9.1, https://cytoscape.org/, accessed on 25 October 2022). The modules in the PPI network were analyzed using the MCODE plugin (Version 1.6.1, https://apps.cytoscape.org/apps/mcode, accessed on 25 October 2022) in Cytoscape software [47].

### 2.7. Transfection of Cultured ASMCs with miRNA Mimics or Inhibitor

ASMCs were transfected with the mimics or inhibitor of specific DE-miRNA, in this case miR-370-5p, as it was identified as a primary DE-miRNA involved in high stretch-induced response in ASMCs. The ASMCs were first seeded in 24-well plates and cultured in Dulbecco’s modified Eagle’s medium with 10% fetal bovine serum at 37 °C under 5% CO_2_ atmosphere. When cultured to ~50% confluence, the cells were transfected using Lipofectamine 3000 (#L3000015, Invitrogen, Carlsbad, CA, USA) with either 50 nM of miR-370-5p mimics (small, chemically modified double-stranded RNAs that mimic endogenous miR-370-5p) or scrambled negative control (NC), and either 50 nM of miR-370-5p inhibitor (chemically modified single-stranded RNA molecules designed to specifically bind to and inhibit endogenous miR-370-5p) or a scrambled NC, respectively, for 48 h according to the manufacturer’s instructions.

### 2.8. Assessment of ATP/cAMP Level in Cultured ASMCs

The ATP level in cultured ASMCs was determined using an ATP determination kit (#S0026, Beyotime Biotechnology, Shanghai, China) [48], following the protocol provided by the manufacturer. In brief, after high-stretch treatment (0.5 Hz, 13%, 72 h) or miR-370-5p mimics/inhibitor transfection (50 nM, 48 h) as described earlier, the cells were first lysed using lysate. The lysis was then centrifuged at 12,000× *g* for 5 min (4 °C), and the supernatant was collected for measurement of intracellular ATP. Briefly, a 10 μL sample of cell culture medium or cell lysis suspension was added to 90 μL reaction solution in a 96-well white bottom assay plate followed by incubation for 15 min at room temperature. Luminescence was monitored by a microplate reader (Tecan Spark, TM10M, Meilen, Switzerland) at 563 nm at room temperature.

The cAMP level in cultured ASMCs was measured using an enzyme-linked immunosorbent assay (ELISA). In brief, after static and high-stretch treatment (0.5 Hz, 13%, 72 h), or miR-370-5p mimics/inhibitor transfection (50 nM, 48 h) as described earlier, the cells were subjected to ultrasonic lysis in the presence of the phosphodiesterase inhibitor IBMX (10^−5^ M) in PBS and centrifuged at 14,000× *g* for 10 min under 4 °C. Then the cAMP enzyme-linked immunosorbent-assay kit (#E-EL-0056c, Elabscience, Houston, TX, USA) was employed following the manufacturer’s protocol. Absorbance at 450 nm was measured using a colorimetric 96-well plate reader (Infinite F50, Tecan, Zürich, Switzerland). Data are expressed as ratios of the optical density of groups treated with DE-miRNA mimics/inhibitor to that of the vehicle control groups.

### 2.9. Quantitative PCR Analysis of mRNA Expression in ASMCs Transfected with miRNA Mimics/Inhibitor

Quantitative PCR (qPCR) analysis was used to further evaluate the effect of up/down-regulation of specific DE-miRNA (in this case miR-370-5p) on the mRNA expression level of genes related to signaling that was most enriched in response to high stretch in cultured ASMCs (in this case AK7 and PDE4D in purine metabolism). Primers associated with these genes were obtained from General Biosystems (Chuzhou, China), as shown in Appendix A. Briefly, total RNA was purified using TRI Reagent RNA Isolation Reagent (#T9424, Sigma-Aldrich, St. Louis, MO, USA), and the extracted RNA was quantified using a Nanodrop 2000 Spectrophotometer (Thermo Scientific, Willmington, DE, USA). And then total RNA (500 ng) was used to generate 1st strand cDNA using the Revert Aid First Strand cDNA Synthesis Kit (#K1622, Thermo Scientific, Willmington, DE, USA). The qPCR experiment was performed with PowerUp SYBR Green Master Mix (#A25742, Applied Biosystems, Foster City, CA, USA) using the StepOne real-time PCR system (Applied Biosystems) at 50 °C for 2 min, 95 °C for 2 min, followed by 40 cycles of 95 °C for 15 s, 55 °C for 15 s, and 72 °C for 60 s. The reaction system (10 µL) contained 1 µL of cDNA in triplicates according to the manufacturer’s instructions. Calibration and normalization were performed using the 2^−∆∆CT^ method, where ∆CT = CT (target gene) − CT (reference gene, GAPDH) and ∆∆CT = ∆CT (experiment groups) − ∆CT (control groups). Fold changes in mRNA expression of different genes were calculated as the ratio of experiment groups to the control groups from the resulting 2^−∆∆CT^ values from three independent experiments.

### 2.10. Statistical Analysis

Statistical analysis was performed by using GraphPad Prism 8.0 (Graph Pad Software, San Diego, CA, USA). Unless stated otherwise, each sample was tested in triplicate, and each experiment for a given study condition was repeated a number of times depending on group size (n). Data were reported as means ± S.E.M. Comparisons of means between 2 groups were performed using an unpaired Student *t*-test. Comparisons of means among three or more groups were performed by one-way analysis of variance (ANOVA) followed by a Post Hoc test (Turkey HSD method) or two-way ANOVA followed by the Tukey HSD method. Differences were considered significant at *p* < 0.05.

## 3. Results

### 3.1. High Stretch-Induced Differentially Expressed mRNAs and miRNAs in ASMCs

The mRNA response of ASMCs to high stretch was evaluated with mRNA sequencing (GSE206435). We defined the differentially expressed mRNAs (DE-mRNAs) in experimental groups as mRNAs with count ≥ 10 that had an absolute log2 (fold change) expression ≥1 together with a *p* < 0.05 when compared with their counterparts in static cells. We found a total of 2858 DE-mRNAs, among which 1466 were downregulated and 1392 were upregulated.

The miRNA response of ASMCs to high stretch was evaluated with miRNA sequencing (GSE206435). Similar to mRNAs, we defined the differentially expressed miRNAs (DE-miRNAs) in experimental groups as miRNAs with a count ≥ 10 that had an absolute log2 (fold change) expression ≥1 together with a *p* < 0.05 when compared with their counterparts in static cells. We found, in total, only 12 DE-miRNAs, among which 7 were upregulated, and 5 were downregulated, as shown in Table 1.

These 12 miRNAs were predicted to target 4464 genes in total, among which 1942 were targeted by the 7 upregulated DE-miRNAs, and 2522 were targeted by the 5 downregulated DE-miRNAs. We further compared the target genes of down/up-regulated DE-miRNAs with those of up/down-regulated DE-mRNAs and found 283 genes that overlapped these two groups (142 overlapped the target genes of down-regulated DE-miRNAs and the genes of up-regulated mRNAs, and 141 overlapped the target genes of up-regulated miRNAs and the genes of down-regulated mRNAs, as shown in Figure 2A and Appendix A). We identified these overlapping genes as target DE-mRNAs, whose expressions were visualized in the volcano plot as shown in Figure 2B. It can be seen that hypoxia up-regulated 1 (HYOU1), activating transcription factor 3 (ATF3), solute carrier family 3 member 2 (SLC3A2), SEL1L adaptor subunit of SYVN1 ubiquitin ligase (SEL1L), fibronectin leucine rich transmembrane protein 2 (FLRT2), KLF transcription factor 15 (KLF15), ATPase plasma membrane Ca^2+^ transporting 1 (ATP2B1), lysosomal thiol reductase (IFI30), MIA SH3 domain ER export factor 2 (MIA2), and MAPK interacting serine/threonine kinase 2 (MKNK2) were the top 10 upregulated target DE-mRNAs, and LBH regulator of WNT signaling pathway (LBH), neuronal regeneration-related protein (NREP), thrombospondin 1 (THBS1), catalase (CAT), guanylate cyclase 1 soluble subunit beta 1 (GUCY1B1), peripheral myelin protein 22 (PMP22), phosphodiesterase 7B (PDE7B), RAB7B member RAS oncogene family (RAB7B), stathmin 1 (STMN1), and elastin (ELN) were the top 10 downregulated target DE-mRNAs in ASMCs cultured with high stretch.

### 3.2. High Stretch-Induced Enrichment of Cellular Functions and Signaling Pathways in Cultured ASMCs

The biological function of these 283 target DE-mRNAs was analyzed by KEGG pathway and GO enrichment analysis using DAVID. As shown in Figure 3A and Appendix A, KEGG pathway analysis indicated that purine metabolism-related signaling, mechanical stress/strain-related signaling, such as cGMP-PKG signaling pathway, MAPK signaling pathway, fluid shear stress and atherosclerosis, amoebiasis, and Ras signaling were all enriched to various extents in ASMCs when cultured with high stretch. However, the top enriched signaling in ASMCs when cultured with high stretch appeared to be related to purine metabolism, which involved 9 target DE-mRNAs, including phosphoribosyl pyrophosphate synthetase 1 (PRPS1), GUCY1B1, phosphodiesterase 4D (PDE4D), ectonucleotide pyrophosphatase/phosphodiesterase 1 (ENPP1), adenylate kinase 4 (AK4), phosphodiesterase 5A (PDE5A), PDE7B, adenylate kinase 4 (AK7), NME/NM23 nucleoside diphosphate kinase 1 (NME1).

The results of GO analysis (Figure 3B and Appendix A) also show that purine metabolism-related biological processes, including regulation of GTPase activity (Rho GTPase activating protein 9 (ARHGAP9), FYVE RhoGEF and PH domain containing 6 (FGD6), RAS p21 protein activator 4B (RASA4B), synaptic Ras GTPase activating protein 1 (SYNGAP1), RAS p21 protein activator 4 (RASA4), chimerin 1 (CHN1), Cbl proto-oncogene B (CBLB)) and ATP metabolic process (ATP binding cassette subfamily C member 6 (ABCC6), ATPase H+ transporting V1 subunit B2 (ATP6V1B2), ENPP1, AK4), as well as purine metabolism-related molecular functions including 3’,5’-cyclic-AMP phosphodiesterase activity (PDE4D, ENPP1, PDE7B), nucleoside diphosphate kinase activity (AK4, AK7, NME1), and 3’,5’-cyclic-nucleotide phosphodiesterase activity (PDE4D, PDE5A, PDE7B), were also enriched. Other events, such as mechanical (basement membrane organization and cell migration) and inflammatory (positive regulation of interleukin-6 (IL-6), tumor necrosis factor-alpha (TNFα), monocyte chemoattractant protein-1 (MCP-1), and bone morphogenetic protein 1 (BMP) production)-related events, were also enriched in ASMCs treated with high stretch.

### 3.3. High Stretch-Induced Enrichments of PPI in Cultured ASMCs

The protein–protein interaction (PPI) network of DE-mRNAs contained 93 nodes and 115 interactions that clustered into three modules, as shown in Figure 4A. The KEGG analysis (Figure 4B and Appendix A) indicates that the 38 target DE-mRNAs in the first cluster of the PPI network were enriched due to high stretch in various functions of the cultured ASMCs related to cellular metabolism, including, in rank-order, purine metabolism (PDE4D, ENPP1, AK4, PDE5A, PDE7B, AK7, NME1), nucleotide metabolism, tryptophan metabolism, metabolism pathways, biosynthesis of cofactors, and thiamine metabolism. The GO analysis (Figure 4C) further indicates that the biological processes and molecular functions enriched in cultured ASMCs due to high stretch include, in rank order, nucleotide diphosphate phosphorylation (AK4, AK7, NME1), ATP metabolic process (ABCC6, ENPP1, AK4), cAMP catabolic process (PDE4D, PDE7B), nucleoside triphosphate biosynthetic process (AK4, AK7), and nucleoside monophosphate phosphorylation (AK4, AK7), 3′,5′-cyclic-cAMP phosphodiesterase activity (PDE4D, ENPP1, PDE7B), nucleoside diphosphate kinase activity (AK4, AK7, NME1), 3′,5′-cyclic-nucleotide phosphodiesterase activity (PDE4D, PDE5A, PDE7B), and adenylate kinase activity (AK4, AK7). All these data indicate that the 7 target DE-mRNAs, including PDE4D, ENPP1, AK4, PDE5A, PDE7B, AK7, and NME1, were the genes most responsive to high stretch in ASMCs. While some of them were also involved in other cell functions, these genes were all involved in regulating purine metabolism. More specifically, PDE5A is involved in guanosine metabolism and the other six DE-mRNAs are involved in adenosine metabolism. Since the second messenger cAMP and ATP are crucial signaling and energy molecules and regulate mechanical and inflammatory phenotypes of ASMCs, the high stretch-induced signaling related to purine metabolism, particularly the cAMP and ATP metabolism in ASMCs, might be key to understanding the regulatory roles as well as the therapeutic potential of miRNAs in VILI [49,50].

### 3.4. High Stretch-Responsive miRNA–mRNA Interactions in Relation to Regulation of cAMP/ATP in Cultured ASMCs

Figure 5A shows seven pairs of possible interactions between the DE-mRNAs targeted by their corresponding DE-miRNAs, namely, PDE4D–miR-370-5p, PDE4D–miR-543, PDE7B–miR-29b-3p, AK4–miR-148a-3p, AK7–miR-370-5p, ENPP1–miR-194-5p, and NME1–miR-146-5p, with respect to their roles in regulating cAMP and ATP metabolism. Since PDE4D and PDE7B both hydrolyze cAMP (cAMP→AMP), the high stretch-increased mRNA expression of PDE4D in association with the downregulated expression of miR-370-5p or miR-543 may result in decreased cAMP concentration in ASMCs, whereas the decreased PDE7B with upregulated miR-29b-3p may result in increased cAMP concentration in ASMCs.

On the other hand, AK4 and AK7 (in mitochondrial and cytosolic, respectively) either efficiently phosphorylates AMP to generate ATP or catalyzes the reversible transfer of the terminal phosphate group between ATP and AMP (AMP→ATP). Therefore, the high stretch-decreased AK4 with upregulated miR-148a-3p may decrease ATP concentration in ASMCs, while the increased AK7 with downregulated miR-370-5p may increase ATP concentration in ASMCs.

In addition, ENPP1 and NME1 either preferentially hydrolyze ATP with the release of pyrophosphate and adenosine polyphosphates (ATP→AMP) or transfer the ATP gamma phosphate to the NDP beta phosphate via a ping-pong mechanism and thus promote the synthesis of nucleoside triphosphates other than ATP (ATP→ADP). Thus, the high stretch-decreased ENPP1 or NME1 with upregulated miR-194-5p or miR-146-5p may all lead to increased ATP concentration in ASMCs.

Figure 5B and 5C show that the intracellular level of cAMP and ATP as measured by ELISA decreased by 20% (*p* = 0.0248) and increased by 60% (*p* = 0.0047), respectively, in ASMCs cultured with high stretch as compared to their counterparts under static condition.

### 3.5. Effect of miR-370-5p on the High Stretch-Induced Variation of cAMP/ATP Concentration in Cultured ASMCs

Among all the above miRNAs, miR-370-5p can target PDE4D and AK7 to regulate the concentration of cAMP and ATP and enrich the purine metabolisms according to the results of KEGG analysis (Appendix A). Therefore, we speculated that miR-370-5p might be the primary miRNA in regulating high stretch-induced variations of cAMP/ATP levels in ASMCs. To verify this speculation, we transferred the cultured ASMCs with miR-370-5p mimics or inhibitor followed by high-stretch treatment. The cell viability of, and transfection efficiency of, miR-370-5p mimics/inhibitor in ASMCs was evaluated with CCK8 assay and qPCR, respectively. The results indicated that the transfection of miR-370-5p mimics (*p* = 0.0966) or inhibitor (*p* = 0.7105) did not significantly affect the viability of the cultured ASMCs (Figure 6A), which was also highly efficient as it increased or decreased the expression level of miR-370-5p by ~280% (*p* = 0.0001) or ~80% (*p* = 0.0421), respectively (Figure 6B). In addition, the mRNA expression of PDE4D (Figure 6C, *p* = 0.0230) and AK7 (Figure 6D, *p* = 0.0139) was decreased by miR-370-5p, respectively. At the same time, the level of cAMP (Figure 6E, *p* = 0.0331) and ATP (Figure 6F, *p* = 0.0036) inside ASMCs was increased and decreased by miR-370-5p, respectively. Consistent with these effects, we found that the miR-370-5p mimics and inhibitor respectively decreased (*p* = 0.0321) and increased (*p* = 0.0251) the basal tone, but neither significantly affected the relaxation of ASMCs (Appendix A). The basal tone and relaxation of ASMCs were evaluated by measuring cell stiffness using optical magnetic twisting cytometry (OMTC) before (basal tone) and after (relaxation) exposure to isoproterenol (ISO, a conventional β-agonists) as described elsewhere [51,52].

More importantly, the presence of miR-370-5p mimics in cultured ASMCs abolished the high stretch-induced decrease in cAMP level (*p* = 0.0212), as shown in Figure 6G, and reversed the increase in the ATP level (*p* = 0.0044), as shown in Figure 6H. These data suggest that the interactions between miR-370-5p and PDE4D, miR-370-5p, and AK7 may be essential in mediating high stretch-induced pathological responses in ASMCs through modulation of the purine metabolism of the cells.

## 4. Discussion

In this study, we first identified, using RNA-Seq 12, DE-miRNAs and 283 target DE-mRNAs that were highly up/down-regulated in ASMCs in response to high stretch. Further bioinformatic analysis, together with PPI network clustering of these DE-mRNAs, indicate that the high stretch-induced prominent change in the cultured ASMCs occurred in the cells’ purine metabolism. More specifically, the cAMP level decreased but the ATP level increased inside ASMCs due to high stretch. At the same time, miR-370-5p expression was decreased and AK7/PDE4D expression was enhanced by high stretch, and the high stretch-induced change in the cAMP and ATP level inside ASMCs could be partially abolished by miR-370-5p mimics. These data suggest that purine metabolism and related miRNA–mRNA interactions may serve as candidate biomarkers and therapeutic targets for ASMCs in high stretch-related diseases. 

Although ASMCs may respond to high stretch by changing the expression, activation, and/or distribution of signaling molecules, the change of expression is relatively easy to evaluate by RNA-Seq. Therefore, with RNA-Seq data, we identified the genes and miRNAs that are differentially expressed in ASMCs when cultured in either high-stretch or static conditions, as a key to understanding the high stretch-induced events and associated regulating miRNA–mRNA interactions in ASMCs. We determined the differentially expressed mRNA and miRNA by analyzing RNA-Seq data with DESeq2, considering its high specificity and true positive rate [29,30]. In addition, we used DAVID, a highly integrated data-mining environment to determine the functional enrichment of the molecules [53]. Based on miRNA–mRNA integrated bioinformatics, we found that purine metabolism was the most responsive event in ASMCs exposed to high stretch.

Purine metabolism has been widely studied for its essential roles in cell behaviors such as cell energy production, proliferation, and survival. In particular, other than serving as building blocks for DNA, purine nucleotides such as ATP are crucial for providing cellular energy, intracellular signaling, and cofactors to promote cell survival and proliferation [49]. Several diseases have also been linked to purine metabolism [54]. In airways, purine is not only involved in the normal energy flow but can also serve as a VILI marker in bronchoalveolar lavage fluid [55,56,57]. For example, lung tissues dissected from rats after exposure to long-term MV show distinct metabolomic profiles indicating enhanced energy production in the lung [58]. Our results of high stretch-induced dramatical responses in purine metabolism, particularly the enhanced ATP level in ASMCs, are consistent with the in vivo finding. When exposed to cyclic high stretch, cells may experience passive exercise and inevitably enhance ATP consumption. ATP could eventually be consumed during these cellular activities, and, hence, the increased probability of ATP synthesis may compensate for the ATP consumption during cyclic high stretch. Therefore, increased ATP may be a feedback mechanism to enhance cellular response to external mechanical stimuli. In addition to functioning as an essential energy molecule, ATP is secreted from various types of lung cells, including ASMCs, when the cells are subjected to mechanical stimulation such as high stretch. The secreted ATP then can mediate the change of basal tone and/or contractility of ASMCs [59]. ATP can also activate purinergic receptors to potently regulate both the physical and chemical behaviors of ASMCs. We did observe that ASMCs exposed to increased levels of ATP became stiffer, more contractile, and pro-inflammatory but not more proliferative. Therefore, the high stretch-enhanced ATP synthesis in ASMCs may be responsible for increased contractility and thus decreased compliance of the airways of mechanically ventilated patients.

Previous studies have shown that miRNAs may change the function of purine metabolism. For example, miR-151a-5p is reported to participate in the regulation of cellular respiration and target cytochrome B for the production of ATP [60]. In this study, we found that ASMCs responded to high stretch with marked changes in interactions between certain miRNAs and their target genes, including miR-370-5p–PDE4D, miR-543–PDE4D, miR-29b-3p–PDE7B, miR-370-5p–AK7, miR-148a-3p–AK4, miR-194-5p–ENPP1, and miR-146-5p–NME1, which may all impact the mitochondrial respiratory function. Among them, miR-370-5p–AK7 is more likely to be involved in ATP synthesis inside ASMCs during high stretch because AK7 is known to function as phosphotransferase in energy homeostasis, and as adenylate kinase to catalyze the production and breakdown of adenine nucleotide in a reversible transfer method [54]. It has also been shown that depletion of AK7 protein will lead to impaired mitochondrial respiratory function, especially the synthesis of ATP [61].

On the other hand, cAMP is a ubiquitous second messenger and its principal function in airways is to mediate relaxation/contraction of ASMCs [50]. For example, the classical bronchodilator, β2-agonist, activates the Gs-coupled β2 adrenergic receptor (β2AR) on ASMCs to activate adenylyl cyclase and, hence, increase the intracellular level of cAMP to mediate relaxation of ASMCs. Acetylcholine released from postganglionic parasympathetic nerves instead inactivates adenylyl cyclase and, hence, lowers the intracellular level of cAMP to mediate contraction of ASMCs. Thus, the intracellular level of cAMP is essential for balancing the contraction and relaxation of ASMCs. In more detail, cAMP is mainly generated by adenylyl cyclase V and VI from cytoplasmic ATP and broken down by intracellular phosphodiesterases. In this study, we found that high stretch decreased intracellular levels of cAMP in association with modulated mRNA expressions of several cAMP-specific 3’,5’-cyclic phosphodiesterases, including PDE4D. Previous studies have widely used optical magnetic twisting cytometry (OMTC) to measure cell stiffness as a surrogate assay in order to assess contraction/relaxation behaviors, such as basal tone and contractility of ASMCs, especially when the cells are cultured with cyclic stretch, exogenous cAMP, and ATP [51,52]. Generally, cyclic stretch and exogenous ATP can enhance basal tone or contractility, but cAMP does the opposite, which has been commonly attributed to the cytoskeletal reorganization and associated Ca^2+^ signaling in the cells. In this study, however, we found that high stretch modulated intracellular levels of cAMP/ATP via miRNA–mRNA, particularly miR-370-5p–PDE4D/AK7 interactions. In addition, we found that miR-370-5p did impact cell stiffness, but not the cell stiffness reduction induced by β-agonist, suggesting that high stretch through miR-370-5p–PDE4D interaction is a mechanical modulator of bronchial airway tone [62]

In addition to the modulation of airway tone through contraction/relaxation, ASMCs also modulate airway inflammation by secreting or expressing a wide array of immunomodulatory mediators in response to extracellular stimuli such as high stretch [63]. For example, mechanical stress can induce more pro-inflammatory mediators including IL-1β, IL-8, IL-6, and TNF-α in the distal air spaces of the lung [64]. In this study, we found that high stretch led to enriched positive regulation of production for IL-6, TNF-α, MCP-1, and BMP-1. These inflammatory factors are potential therapeutic targets for VILI since they are upstream targets in response to stretch and initiate many pathophysiological processes linked to VILI, including barrier permeability, alveolar fluid clearance, and apoptosis [65,66]. Although ASMCs share with canonical immune cells the same mechanisms to secrete cytokines and chemokines, the unique miRNA–mRNA interactions revealed in this study for regulating inflammatory factors in ASMCs may provide novel therapeutic targets for VILI. The demonstration of high stretch-modulated cAMP and ATP levels and related purine metabolism inside ASMCs via the miR-370-5p–PDE4D/AK7 pathway also warrants the exploration of these effects in other lung cells that are also important in VILI, such as airway epithelial cells and macrophages. For example, cAMP triggers Na^+^ absorption, anti-inflammation, mucociliary clearance of the airway epithelium, and has been recognized as a potential therapeutic target of cystic fibrosis [34,67]. Therefore, when other types of lung cells were targeted, the miR-370-5–PDE4D/AK7 pathway may have had other pharmacological functions.

To our knowledge, this is the first report of differential responses of miRNAs and mRNAs in ASMCs when exposed to high stretch using a whole genome-wide approach, which surprisingly linked high stretch to the purine metabolism, especially cAMP and ATP synthesis in ASMCs. However, there are limitations to this work. For instance, the purine metabolism modulated by miR-370-5p–PDE4D/AK7 interactions to regulate cAMP/ATP levels inside ASMCs during high stretch was identified solely based on target prediction, statistical evidence, and mRNA evaluation. The effects of high stretch on PDE4D and AK7 in ASMCs are yet to be experimentally verified at the protein level using methods such as dual-luciferase reporter assay and protein evaluation. It is also known that the effect of cyclic stretch on ASMCs is amplitude-dependent. Although previous studies have already shown that high stretch can cause pathological effects on ASMCs, such as enhanced ATP secretion and disrupted alveolar epithelial barrier integrity [59,68], it is unclear whether cyclic stretch at low tidal volume or gradient amplitude would incrementally change ASMCs as they proceed toward pathological remodeling so that more pathogenic signals/pathways may be identified. Furthermore, it remains unknown whether the high-stretch modulation of intracellular levels of cAMP and ATP depends on specific membrane transporters, such as ATP-binding cassettes, and how cAMP and ATP coordinate to regulate downstream signals such as AMP-activated protein kinase activity. Finally, the in vitro findings need to be investigated with in vivo animal models and ultimately human subjects to further confirm their pharmacological and therapeutic implications in relation to the treatment of VILI.

## 5. Conclusions

The present study revealed that purine metabolism regulated through miRNA–mRNA interactions was a prominent cellular response of ASMCs to high stretch in association with significant down/up-regulation of intracellular cAMP/ATP levels. Furthermore, miR-370-5p for transcription regulation of PDE4D and AK7 was the major regulatory molecule responsible for high stretch-induced changes related to purine metabolism in ASMCs. Taken together, the link established between the physiological functions, such as purine metabolism and related miRNA–mRNA interactions, in ASMCs during high stretch may help discover potential new biomarkers and targets for diagnosis and treatment of respiratory diseases associated with MV, such as VILI.

## Figures and Tables

**Figure 1 cells-13-00110-f001:**
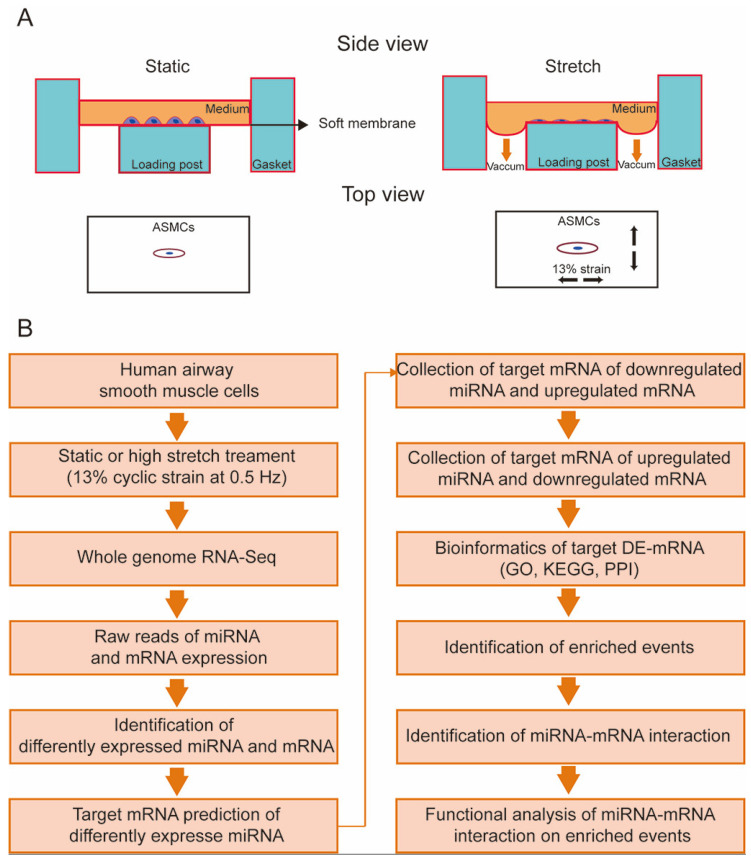
Schematic diagram of the experimental protocol to identify high stretch-responsive events and the underlying regulatory miRNA–mRNA interactions in human airway smooth muscle cells (ASMCs). (**A**) The control and experimental ASMCs were cultured on an elastic membrane in either static (left) or high-stretch (right, 13% cyclic strain, 0.5 Hz) conditions, respectively. (**B**) Flow chart of high-stretch treatment, whole genome-wide RNA-Seq, bioinformatics analysis, and functional analysis to identify high stretch-induced enriched events and miRNA–mRNA interactions in cultured ASMCs.

**Figure 2 cells-13-00110-f002:**
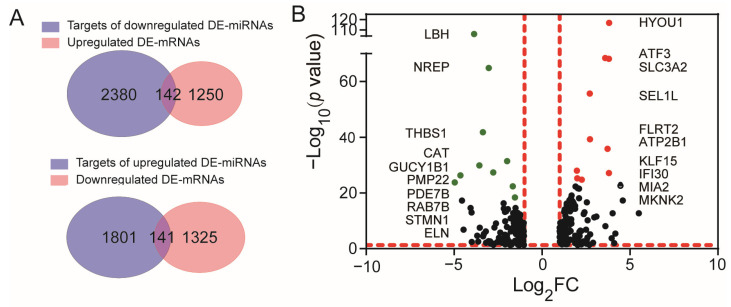
Identification of target differentially expressed mRNAs (DE-mRNAs) in ASMCs cultured under static or high-stretch conditions. (**A**) Overlapped genes between targets of down-regulated DE-miRNAs and up-regulated DE-mRNAs (up panel) and overlapped genes between targets of up-regulated DE-miRNAs and down-regulated DE-mRNAs (down panel). (**B**) Volcano plot of 283 target DE-mRNAs. Black dots indicate the genes that were differently expressed in ASMCs cultured in high-stretch groups compared to static groups with *p* < 0.01 and |log_2_FC| > 2. Red dots indicate the top 10 upregulated mRNAs and green dots indicate the top 10 down-regulated mRNAs.

**Figure 3 cells-13-00110-f003:**
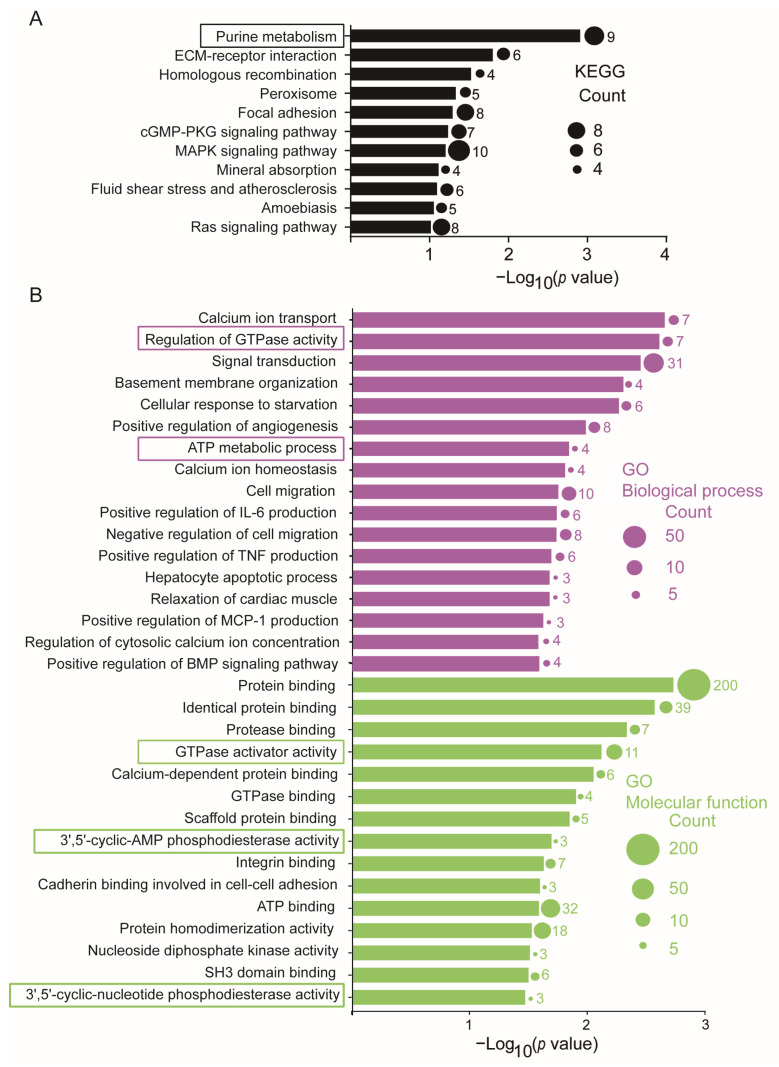
Purine metabolism was enriched in the KEGG pathway and GO enrichment analysis of the 283 target DE-mRNAs. (**A**) Significant enrichment of the KEGG pathway of the 283 target DE-mRNAs. (**B**) Significant enrichment GO of the 283 target DE-mRNAs. Purple bars stand for the BP, and green bars stand for the MF. Box indicates that these events were related to purine metabolism. The circle indicates the number of enriched genes in each term.

**Figure 4 cells-13-00110-f004:**
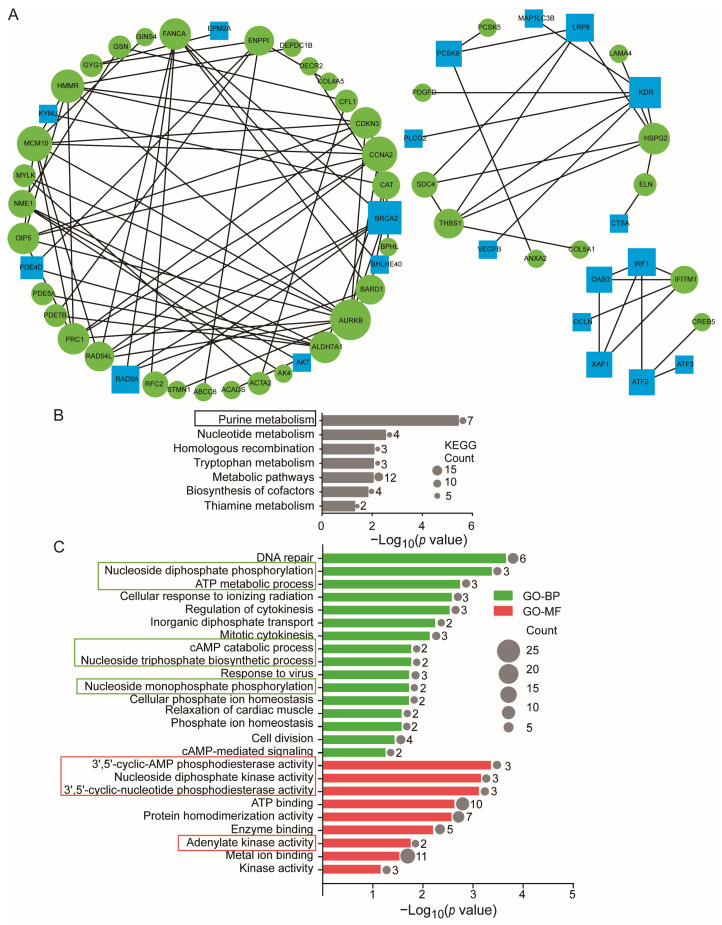
Purine metabolism was enriched in the first module of the PPI network. (**A**) The PPI network construction and module analysis. The blue nodes represent the up-regulated target DE-mRNAs. The green nodes represent the down-regulated target DE-mRNAs. The size of the nodes indicates the degree of the target DE-mRNAs, and the lines represent the interaction between gene-encoded proteins. (**B**) Significant enrichment of the KEGG pathway of the 38 DE-mRNAs in the first module of the PPI network. Box indicates that these events were related to purine metabolism. (**C**) Significant enrichment GO of the 38 DE-mRNAs in the first module of the PPI network. Green bars stand for the BP, and red bars stand for the MF. Box indicates that these events were related to purine metabolism. The circle indicates the number of enriched genes in each term.

**Figure 5 cells-13-00110-f005:**
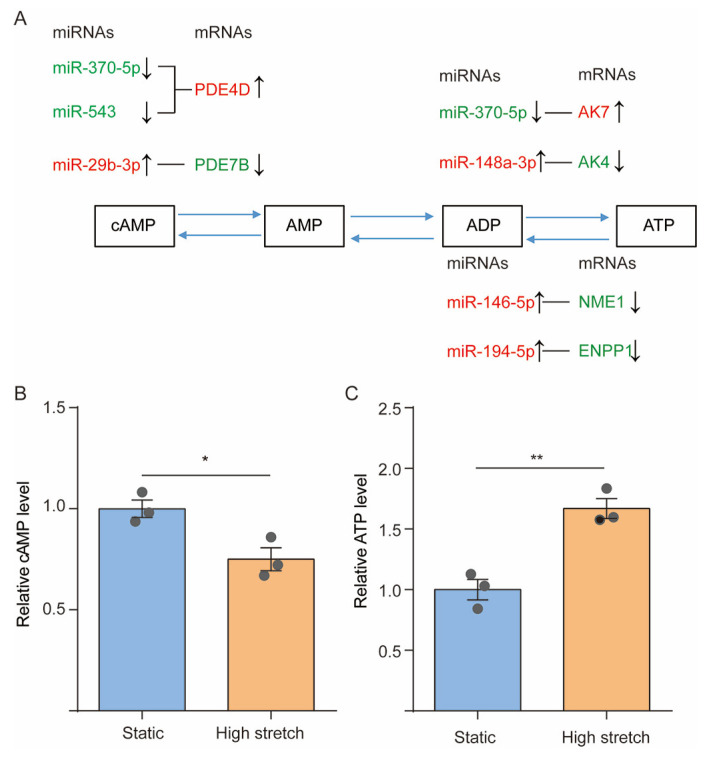
High stretch modulated purine metabolism. (**A**) The miRNA–mRNA interactions related to purine metabolism response to high stretch of ASMCs. The red and green colors represent up-regulation and down-regulation, respectively. The bar graph shows the relative cAMP (**B**) and ATP (**C**) levels inside ASMCs cultured in static or high-stretch conditions. Each experiment was repeated three times (*n* = 3). Data are presented as means ± S.E.M. * *p* < 0.05 and ** *p* < 0.01 compared to static groups.

**Figure 6 cells-13-00110-f006:**
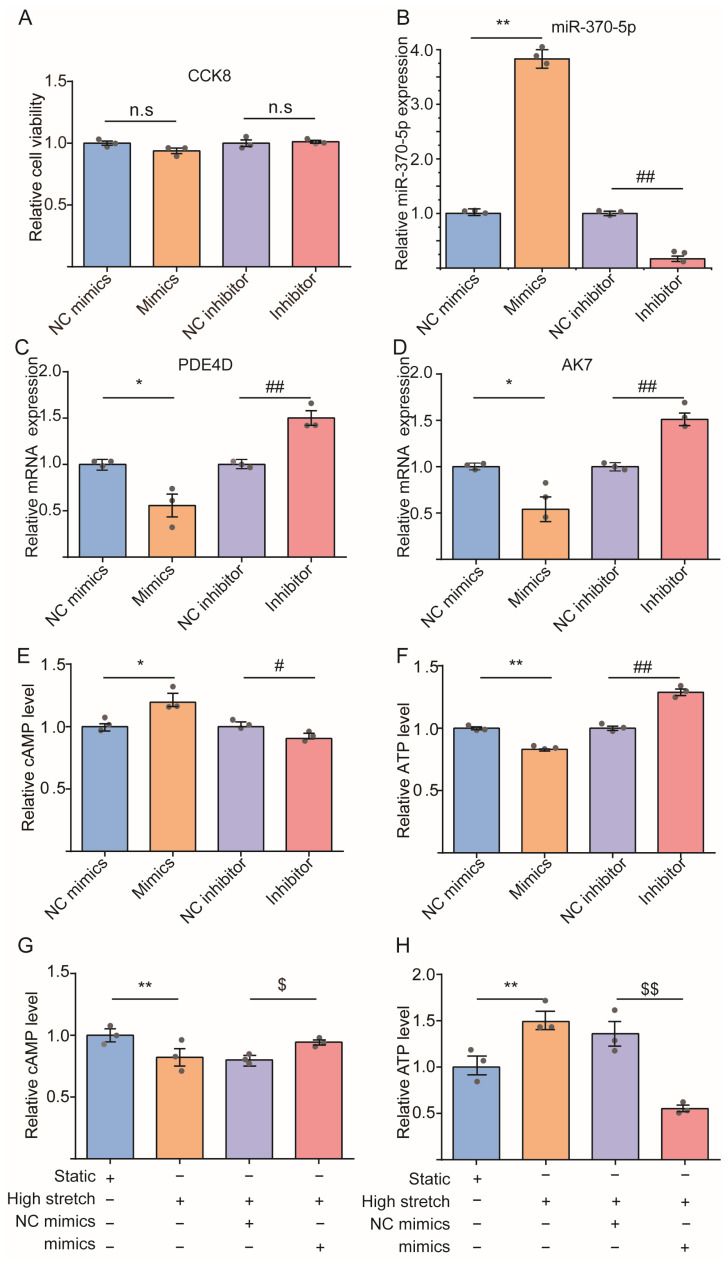
miR-370-5p partially reversed high stretch-induced change in cAMP and ATP levels inside ASMCs cultured in static or high-stretch conditions. (**A**) The cellular activity of ASMCs treated with miR-370-5p mimics or inhibitor. (**B**) Relative expression levels of miR-370-5p in ASMCs treated with miR-370-5p mimics or inhibitor. (**C**,**D**) The mRNA expression of PDE4D, AK7 of ASMCs treated with miR-370-5p mimics or inhibitor, respectively. (**E**,**F**) The relative cAMP, ATP level inside ASMCs cultured with miR-370-5p mimics or inhibitor, respectively. (**G**,**H**) The relative cAMP, ATP level inside ASMCs cultured in static or high-stretch conditions with or without miR-370-5p mimics. Each experiment was repeated three times (*n* = 3). Data are presented as means ± S.E.M. Markers denote not significant (n.s) or significant difference from control conditions on NC mimics or static group (* *p* < 0.05), NC inhibitor (^#^ *p* < 0.05), high-stretch group with NC mimics (^$^ *p* < 0.05), ** *p* < 0.01, ^##^
*p* < 0.01, ^$$^
*p* < 0.01.

**Table 1 cells-13-00110-t001:** High stretch-induced differentially expressed miRNAs (DE-miRNAs) in cultured ASMCs (stretch/static, count ≥ 10; *p* < 0.05; Fold change > 2).

No.	DE-miRNA	Log_2_FC(Stretch/Static)	*p* _adjust_	Regulate
1	miR-12136	2.25	3.07 × 10^−62^	up
2	miR-192-5p	1.40	3.22 × 10^−15^	up
3	miR-146a-5p	1.29	5.34 × 10^−3^	up
4	miR-194-5p	1.18	1.31 × 10^−17^	up
5	miR-29b-3p	1.16	2.23 × 10^−7^	up
6	miR-148a-3p	1.16	7.06 × 10^−20^	up
7	miR-137-3p	1.10	8.46 × 10^−9^	up
8	miR-370-5p	−1.01	2.93 × 10^−4^	down
9	miR-27b-5p	−1.23	1.63 × 10^−11^	down
10	miR-543	−1.30	2.80 × 10^−19^	down
11	miR-485-3p	−1.47	9.13 × 10^−19^	down
12	miR-335-3p	−2.16	4.85 × 10^−7^	down

## Data Availability

The raw data supporting the conclusions of this article will be made available by the authors, without undue reservation.

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
