# Peer review of "High Stretch Modulates cAMP/ATP Level in Association with Purine Metabolism via miRNA–mRNA Interactions in Cultured Human Airway Smooth Muscle Cells"

_cells, 2024, doi:10.3390/cells13020110_

Round 1
Reviewer 1 Report
Comments and Suggestions for Authors
This is a highly comprehensive [and detailed] study exploring the non-genetic mechanisms (i.e., post-transcriptional gene regulation by miRs) in isolated human airway smooth muscle cells in response to imposed high stretch (i.e., 72h, 13% cyclic strain) that is suggested to mimic mechanical ventilation (high tidal volume) modality used in support of patients with ARDS and associated ventilation-induced lung injury. Using miR and mRNA sequencing, the authors showed 12 significant differentially expressed miRs in stretch vs static conditions and found 283 potential mRNA targets. Using KEGG and GO enrichment analyses, the authors showed an enrichment of purine-metabolism-related signaling in the dataset, and narrowed down to a number of miR-mRNA interactions that may influence intracellular levels of cAMP and ATP. They then focused on miR-370-5p (down) and its discordant effects on PDE4D mRNA (increased) and AK7 mRNA (decreased) in response to stretch, resulting in decreased cAMP and increased ATP, respectively. Finally, they showed that changes in cAMP and ATP levels in response to stretch can be partially recovered by transfecting human airway smooth muscle cells with miR-370-5p mimic. The manuscript is well-written and concise, the experiments are carefully designed, and the findings offer new conceptual insights into pathological mechanisms of ventilation-induced lung injury at the level of smooth muscle embedded in the airways. I have a few suggestions.
1. Abstract: Please correct "382" DE-mRNAs to "283".
2. Materials and Methods: I would recommend significantly shortening the section background as many of the methods/analyses are well-established and reported.
3. Fig. 5 and Fig 6: Please state the n=technical replicates and how many repeated measurements. Also, please plot individual data on the bar graph.
4. Does cyclic stretch and the observed changes in the intracellular cAMP (20% decreased) and ATP (60% increased) levels effect basal tone or the contractility of human airway smooth muscle cells? Or, the viability of the cells? On that point, does cyclic stretch effect AMPK activity?
5. In Fig. 6C, miR-370-5p mimic significantly decreased PDE4D mRNA levels. Have your confirmed PDE4D protein levels and changes in the intracellular cAMP. Also for AK7 protein and ATP levels? Please correct "PED4D" to "PDE4D" in Fig. 6C
6. What are the effects of miR-370-5p mimic and/or inhibitor on the basal tone or relaxation in response to b-agonists?
7. It is now established that cAMP is egressed or released from human airway smooth muscle cells. Can the authors speculate or discuss the effects of cyclic stretch on cAMP secretion (homeostatic regulation of cAMP)?
8. The authors studied static vs cyclic high stretch. It would have been interesting to include a low tidal volume stretch as done in the clinic in order to identify the pathogenic signals/pathways.
9. Discussion: it would be helpful to put the findings in context of the integrated tissue responses to stretch, including other immune and resident lung cell responses.
Reviewer 2 Report
Comments and Suggestions for Authors
Some of the justification/explanation for the experiments should be in the Discussion, not Methods or Results as it disrupts the flow, making the paper difficult to follow at times. Sample size unclear. Please state the exact P-values. Below are just some major comments:
Title:
1) The title is too long, perhaps simplify or shorten it a little.
Abstract:
1) Were miR-370-5p or PDE4D/AK7 previously linked to VILI or any other lung diseases? Perhaps one line info about them would be help tie your results.
Introduction:
1) The Introduction is too long. The 4th paragraph can be shortened by deleting what this study was found as it will be mentioned in the Discussion again.
2) COVID-19 was used as an example for VILI, which is fine. However, it should be mentioned as such, an example, as this study is not specifically looking at COVID-19 induced VILI. The way it is written now has put too much emphasis on COVID-19.
3) Information and advantages of RNA-seq and gene-annotation enrichment analyses can be shortened.
4) Line 39: 10 references for one statement is a bit excessive.
Methods:
1) Never start a sentence with an abbreviation.
2) What is the passage number for the ASMCs used in the experiments
3) Figure 1B: isn’t the Control underwent the same process?
4) Were the experiments repeated?
5) Section 2.3: don’t have to elaborate background information of RNA-Seq. For example, the 1st and 2nd paragraphs are unnecessary.
6) Line 149: again, didn’t the Control undergo the same process?
7) Section 2.4: again, for the 1st paragraph, keep the explanation to the Discussion.
8) Section 2.5: the 2nd paragraph can be shortened and moved to Discussion.
9) Line 259: The ‘m’ for ‘manufacturer’ doesn’t have to be capital letter. Check for others.
10) The sequences for the miRs mimics can be in a Table.
11) Section 2.8: didn’t the control groups underwent the same process
12) Section 2.9: What are the housekeeping/reference genes?
13) 2.10: P-value to be considered significant must be stated.
Results:
1) Line 323: P-value must be <0.05, not ≤0.05 to be considered significant. Can the authors please review their results again if any results with P=0.05 was considered significant.
2) You need to define the name of the genes when you first mentioned it, cannot just use the abbreviation.
3) Line 455: what is the exact P-value?
4) Section 3.5: please insert the exact P-values in-text.
5) Certain information in the Results should be moved to Discussion. Results section should only be about the results, not justification which should be in the Discussion.
6) For the figures with bar graph, can you please change it to a dot plot instead so we can see the variability.
Discussion:
1) Can you please discuss the results in relation to diseases related process.
Round 2
Reviewer 2 Report
Comments and Suggestions for Authors
There has been improvement to the manuscript and the authors have taken the feedback well. A few minor comments to address:
Methods:
1) Have you check if using the ASMCs at passage 10 is still good to use? As the cells may have lost their multipotency in passage 10.
Results:
1) Figure 5: should have * indicate in the bar graphs.
2) Figure 6: missing the symbols to indicate differences. The exact P-value is important for in-text, for the figures, you just need to use symbols to indicate the differences.
Discussion:
1) 2nd paragraph: this paragraph can really be shortened further. A lot of background information about the methods can be simplified as it is not the major point of this study.
2) Line 512: what does “egress” means here?
3) Do not start a sentence with an abbreviation, check your manuscript again. Found one in the Discussion.
4) Line 516-518: please rewrite this sentence, don’t start with “actually”.
5) Line 575-576: can you please rewrite this sentence, doesn’t read well.
Comments on the Quality of English LanguageOk.
